# Integrative analysis of the metabolome and transcriptome provides insights into the mechanisms of flavonoid biosynthesis in *Polygonatum*

Xiaolin Wan[1,2☯], Qiang Xiao[1,2☯]*

1 Hubei Key Laboratory of Biological Resources Protection and Utilization (Hubei Minzu University), Enshi, China, 2 Hubei Key Laboratory of Selenium Resource Research and Biological Application (Hubei Minzu University), Enshi, China

☯ These authors contributed equally to this work.

* 1992022@hbmzu.edu.cn

**Data Availability Statement:** All relevant data are within the manuscript and its Supporting Information files.

## Abstract

A noteworthy group of culinary and medicinal plants is *Polygonatum* species. They are known for their abundant flavonoid compound-rich rhizomes, which have antioxidative and anticancer activities. Using *Polygonatum sibiricum* Red (SXHZ) and *Polygonatum kingianum* var. *grandifolium* (HBES), we conducted transcriptome and metabolomic investigations to look into the molecular processes that control the manufacture of these flavonoids in *Polygonatum* plants. Seven distinct flavonoid metabolites were identified by the analytical data, with phloretin exhibiting a notable differential expression in the biosynthetic pathway. 30 genes with differential expression were found in both plants after further investigation, five of which are members of the transcription factor family associated with MBW. Thus, we suggest that Phloretin and the genes belonging to the MYB-related transcription factor family play a crucial role in controlling the flavonoid biosynthesis pathway in *Polygonatum*. This work lays the groundwork for a deeper comprehension of the biosynthesis and metabolic processes of flavonoids in *Polygonatum*, serving as an invaluable resource for the development of the *polygonatum*-related pharmaceutical industries as well as for the future breeding of *Polygonatum* plants with higher flavonoid content.

## Introduction

*Polygonatum* plants are medicinal perennial herbs of the lily family, primarily grown in the understory [1, 2]. In traditional Chinese medicine, the dried rhizome of *Polygonatum* is widely used to treat influenza, dizziness, cough, diabetes, indigestion, loss of appetite, back pain, and respiratory diseases [3, 4]. *Polygonatum* contains various active components, such as polysaccharides, steroid saponins, flavonoids, triterpene saponins, lignin, alkaloids, fatty acids, and coumarins [5]. Among these, the flavonoid content in *Polygonatum* is rich and has significant medicinal value [6]. Pharmacological studies have shown that the flavonoids in *Polygonatum*

**Funding:** This study was jointly funded by the National Natural Science Foundation of China [NSFC, grant number 31260057, QX), Natural Science Foundation of Hubei Province (Joint Fund) [NSFHP, grant number 2023AFD077, QX], The Open Fund of Hubei Key Laboratory of Biological Resources Protection and Utilization (Hubei Minzu University) [OHBP, grant number KYPT012403, QX], The Open Fund of Hubei Key Laboratory of Selenium Resource Research and Biological Application (Hubei Minzu University) [OHSR, QX], and Major Special Project of Technological Innovation of Hubei Provincial Science and Technology Department [MHT, grant number 2019ACA120, QX]. The funders had no role in study design, data collection and analysis, decision to publish, or preparation of the manuscript.

**Competing interests:** The authors have declared that no competing interests exist.

possess various beneficial effects, including anticancer [7], antioxidant [8], anti-atherosclerosis [9], antibacterial [10], hypoglycemic, and anti-hyperlipidemic activities [11]. Additionally, studies indicate that *Polygonatum sibiricum* Red (SXHZ) rhizomes are fiber-free and can be consumed directly. *Polygonatum kingianum* var. *grandifolium* (HBES), a variant of *Polygonatum kingianum* Coll.et Hemsl exhibits high biological yield and strong adaptability [12]. However, research on the differences in synthesizing secondary metabolites between these two rhizomes still needs to be completed. Therefore, investigating the regulatory mechanisms of flavonoid synthesis in both types of *Polygonatum* is of significant theoretical and practical value for cultivating varieties with high flavonoid content.

Transcription factors (TFs) are essential in regulating gene expression, influencing the expression levels of specific genes by promoting or inhibiting their transcription [13]. Internal and external cellular environments regulate their activity and selective binding. This regulatory mechanism is crucial in various biological processes, such as cell differentiation, development, and stress responses [14]. Studies have shown that transcription factors play a regulatory role in the biosynthesis of flavonoids, which may be mediated by one or more TFs or achieved through specific TF complexes such as the MYB-bHLH-WD40 complex [15]. For example, HuWRKY40, a WRKY transcription factor family member, can promote flavonoid synthesis and improve the fruit quality of *Hylocereus undatus* during storage [16]. In *Erigeron breviscapus*, it has been found that the R2R3-MYB family member EbMYBP1 can directly bind to the promoters of flavonoid-related genes such as FLS, F3H, CHS, and CHI, thereby activating their transcription and promoting the biosynthesis of flavonoids [17]. In addition to transcription factors, enzymes also play essential roles in synthesizing flavonoids. For example, chalcone synthase (CHS) can direct significant carbon fluxes for flavonoid biosynthesis and catalyze downstream enzymatic reactions to generate a wide range of flavonoids [18]. Moreover, further enzymatic modifications contribute to flavonoids' structural and functional diversity [19].

Currently, the biosynthesis of flavonoid compounds in the rhizomes of different *Polygonatum* species has been explored to some extent. However, the regulatory mechanisms and the specific roles of the involved transcription factor families still need to be better understood. Therefore, enhancing the flavonoid content in *Polygonatum* rhizomes through genetic modification and cultivating high-flavonoid varieties holds promising application prospects. This study focuses on the rhizomes of HBES and SXHZ as research materials, aiming to explore the transcriptional and metabolic mechanisms in the flavonoid biosynthesis of *Polygonatum* species. The primary objectives of this research include (i) constructing an mRNA library and identifying related metabolites; (ii) screening differentially expressed genes (DEGs) and differentially expressed metabolites (DEMs) related to flavonoid biosynthesis and metabolism; and (iii) performing correlation analysis between DEGs and DEMs. These findings will deepen the understanding of the molecular and metabolic regulatory mechanisms of flavonoid biosynthesis and provide valuable insights for the breeding and cultivation of high-quality *Polygonatum* varieties and their applications in the pharmaceutical industry.

## Materials and methods

### Plant materials

The test materials consisted of three-year-old rhizomes of *Polygonatum kingianum* var. *grandifolium* (HBES) and *Polygonatum sibiricum* Red (SXHZ). The *Polygonatum* plants were cultivated at the experimental site of the College of Forestry and Horticulture, Hubei Minzu University. The underlying soil was composed of vermiculite and peat soil in a ratio of 8:1. Each set of *Polygonatum* rhizome sequencing samples was subjected to three biological replicates. The collected *Polygonatum* rhizomes were cut into smaller pieces, combined, and placed

into sterile centrifuge tubes. They were then rapidly frozen using liquid nitrogen and stored at a temperature of -80°C until they were ready to be used.

## Metabolite extraction and analysis

The primary system for data acquisition was comprised of ultra-performance liquid chromatography (UPLC) (SHIMADZU Nexera X2) and tandem mass spectrometry (MS/MS) (Applied Biosystems 4500 QTRAP).

The liquid phase conditions were as follows: (i) The chromatographic column used was an Agilent SB-C18 1.8 μm, with dimensions of 2.1 mm * 100 mm; (ii) The mobile phases consisted of ultrapure water (with an addition of 0.1% formic acid) for phase A and acetonitrile (also with 0.1% formic acid added) for phase B; (iii) The B-phase ratio was initiated at 5% at 0.00 min, then linearly increased to 95% over a duration of 9.00 min, and maintained at this level for 1 min. From 10.00 to 11.10 min, the B-phase ratio was reduced to 5% and equilibrated at this value for 14 min; (iv) The flow rate was set at 0.35 mL/min with column temperature at 40°C and an injection volume of 4 μL. The UPLC-MS/MS was conducted by Metware Biotech Ltd (Wuhan, China). Metabolomics data were garnered in both electrospray ionization negative (ESI-) and positive (ESIIC) modes. The ion spray voltage was set at -4500 V for ESI- and 5500 V for ESIC; the ion source gas I (GSI), gas II (GSII), and curtain gas (CUR) were set to 50, 60, and 25 psi, respectively. The collision-induced ionization parameter was set to high, and the temperature of the electrospray ionization source (ESI) was 550°C. Furthermore, the metabolome data underwent principal component analysis (PCA), partial least squares discriminant analysis (PLS-DA), orthogonal partial least squares discriminant analysis (OPLS-DA), differential metabolite expression analysis, and Kyoto Encyclopedia of Genes and Genomes (KEGG) pathway analysis.

## RNA extraction and sequencing analysis

Total RNA was extracted from *Polygonatum* rhizomes using the Plant RNA Kit (200) R6827-02 (OMEGA, USA). The integrity and possible contamination of the RNA were analyzed by agarose gel electrophoresis. The purity of RNA was determined using the Qsep400 High-Throughput Nucleic Acid Protein Analysis System (Ginkgo BioWorks, China). The RNA concentration was measured with a Qubit 4.0 fluorometer (Thermo Fisher Scientific, USA). The RNA integrity was assessed using an Agilent 2100 Bioanalyzer. After RNA quality checks, cDNA libraries were constructed using an automated library construction workstation (MGI Tech, China). Subsequently, sequencing was performed on the Illumina platform. After obtaining clean reads, Trinity [20] was used to splice the clean reads. After splicing was completed, the longest Cluster sequence obtained by hierarchical clustering of Corset [21] was used as the unigene for subsequent analysis.

The sequences of the unigene were analyzed using DIAMOND [22] BLASTX software, and compared with databases including KEGG, National Center for Biotechnology Information (NCBI) non-redundant protein sequences (Nr), Swiss-Prot, Gene Ontology (GO), Clusters of Orthologous Groups of proteins (COG)/euKaryotic Ortholog Groups (KOG), various new documentation files, and the TrEMBL database. Subsequently, these predicted sequences were compared with the Protein family (Pfam) database using HMMER software, providing annotation information for the unigene. Gene expression levels were assessed using Fragments Per Kilobase of transcript per Million fragments mapped (FPKM) and DEGs were defined.

### Gene expression and differential gene analysis

The basic process of gene annotation encompasses protein function annotation, pathway annotation, COG/KOG function annotation, and GO annotation. To annotate individual genes, we utilized the BLASTx program, setting an E-value threshold of 1e-5, in order to compare them with the Nr database, Swiss-Prot database, KEGG database, and COG/KOG database. Subsequently, the TFs are identified by hmmscan comparison using the defined TFs families and rules in the database. A differential gene expression analysis was conducted between two distinct groups using the DESeq2 software. Genes with a false discovery rate (FDR) below 0.05 and a fold change (FC) $\geq$ 2 were considered as DEGs.

### Transcriptome and metabolome association analysis

To identify the critical genes associated with flavonoid synthesis, a comprehensive analysis was undertaken, which involved selecting DEGs and contrasting the differences in flavonoid accumulation between HBES and SXHZ. We used the toolkit in the Rstudio 3.2.2 software to complete the analysis of the relevant data and plot the graphs.

### Quantitative real-time PCR (qRT–PCR) validation

To verify the accuracy of the transcriptome data, we randomly selected 20 gene sequences for primer design and performed qRT-PCR analysis. We extracted RNA using ComWin Biotech (Beijing, China) Plant All-in-One RNA Extraction Kit, and reverse transcribed RNA into cDNA using RTIII All-in-One Mix with dsDNase Reverse Transcription Kit from Monad Biotech (Wuhan, China). primers were synthesized by Sangon Biotech (Sangon, Shanghai, China). The experiment was conducted using an ABI7500 real-time fluorescence quantitative PCR apparatus. The real-time quantitative PCR reaction system consisted of 20 μL, comprising 1 μL of cDNA, 8 μL of RNase-free water, 10 μL of 2X SGExcel FastSYBR Master, and 0.5 μL of each forward and reverse primers. The reaction program involved a pre-denaturation step at 95˚C for 3 minutes, followed by 40 cycles of denaturation at 95˚C for 5 seconds and annealing at 60˚C for 20 seconds. The annealing/extension step should be performed at a temperature of 60˚C for a duration of 20 seconds. The *ubiquitin* was selected as a reference gene: forward primer: 5'- GGACCCAGAAGTACGCAATG-3', reverse primer: 3'- AATTACCAGGGATACAGCA CC-5' [23]. Three technical replicates were prepared for each extract, and the quantitative results were analyzed using the $2^{-\Delta\Delta CT}$ method [24]. The complete list of primers may be seen in S1 Table.

## Results

### Metabolite analysis of two *Polygonatum* rhizomes

We found a total of 652 compounds by detecting both the qualitative and quantitative aspects of metabolites in the rhizomes of both *Polygonatum* species. Principal component analysis (PCA) demonstrated notable disparities in the overall metabolite composition between the HBES and SXHZ sample groups, but the variations within the samples within each group were minimal (Fig 1A). In addition, we utilized latent structure orthogonal projection discriminant analysis (PLS-DA), a sophisticated multivariate statistical technique, to accurately differentiate between group variations and aid in the detection of metabolites that were expressed differently (DEMs) (Fig 1B). Subsequently, we employed orthogonal partial least squares discriminant analysis (OPLS-DA) to assess the model and conduct screening analysis of differential expression maps (DEMs) (Fig 1D). In order to mitigate the issue of overfitting in multivariate statistical analysis, we conducted model validation. The model's reliability was verified by

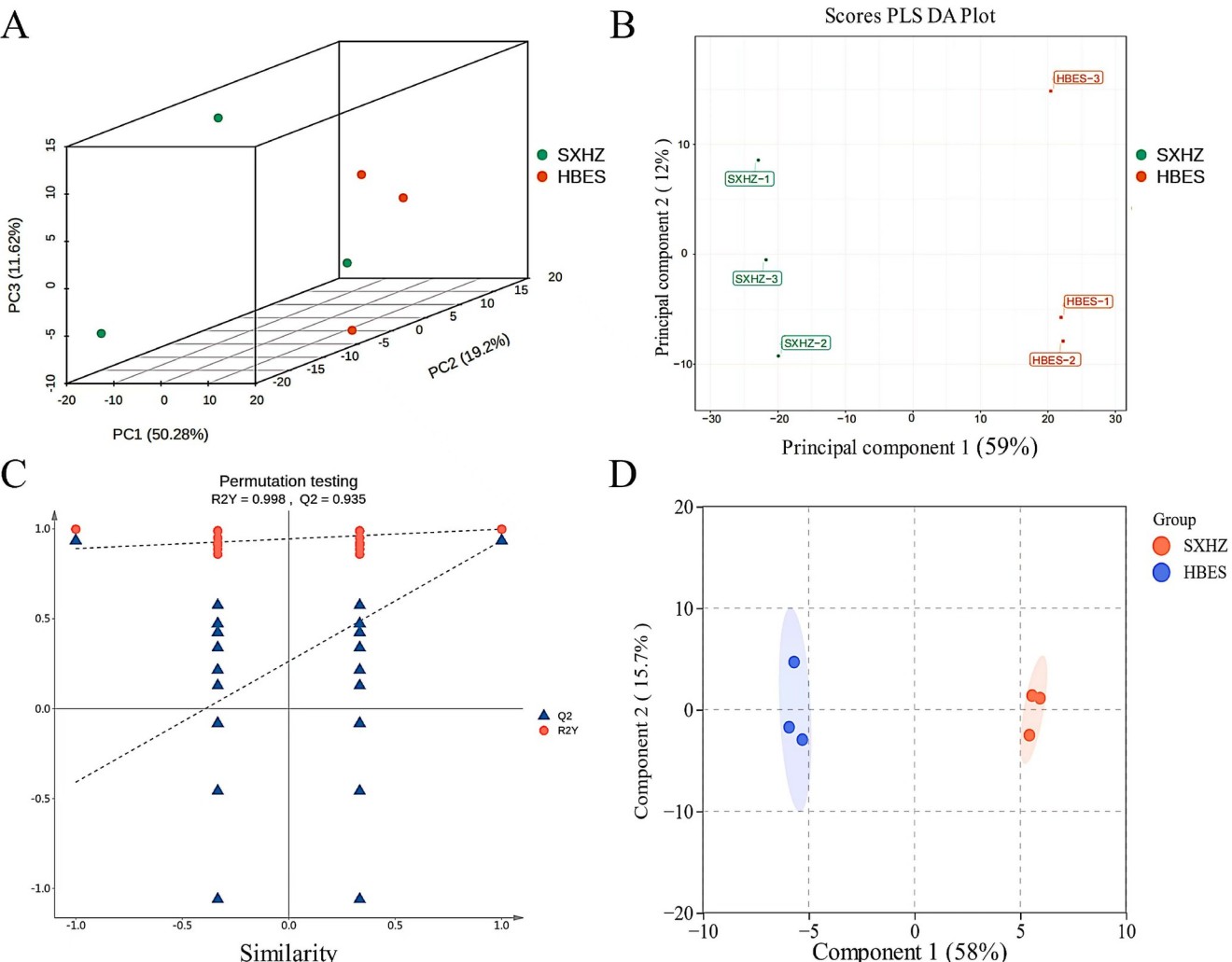

**Fig 1. HBES and SXHZ multivariate statistical analyses.** A: overall sample PCA plot; B: PLS-DA plot for latent structures; C: OPLS-DA validation plot; D: OPLS-DA score plot.

cross-validation and the replacement test, resulting in R2Y = 0.998 and Q2 = 0.935 (Fig 1C). Hence, the findings of our investigation are dependable.

## Analysis of DEMs and enrichment of metabolic pathways

The data from the DEMs analysis revealed that there were a total of 238 DEMs in the comparison between SXHZ and HBES. Among these, 121 genes exhibited up-regulated expression whereas 117 genes showed down-regulated expression (Fig 2A). The KEGG metabolic pathway annotations classify pathways into three main categories: metabolism, genetic information processing, and environmental information processing. Fig 2B depicts metabolic pathways including ten or more DEMs. The metabolic pathway category stands out as having the highest number of DEMs, totaling 175. By examining the top 20 DEMs that exhibited the most significant variations in *Polygonatum* rhizome development, we discovered that succinic acid had the highest concentration, followed by N-acetyl-L-glutamine (Fig 2C).

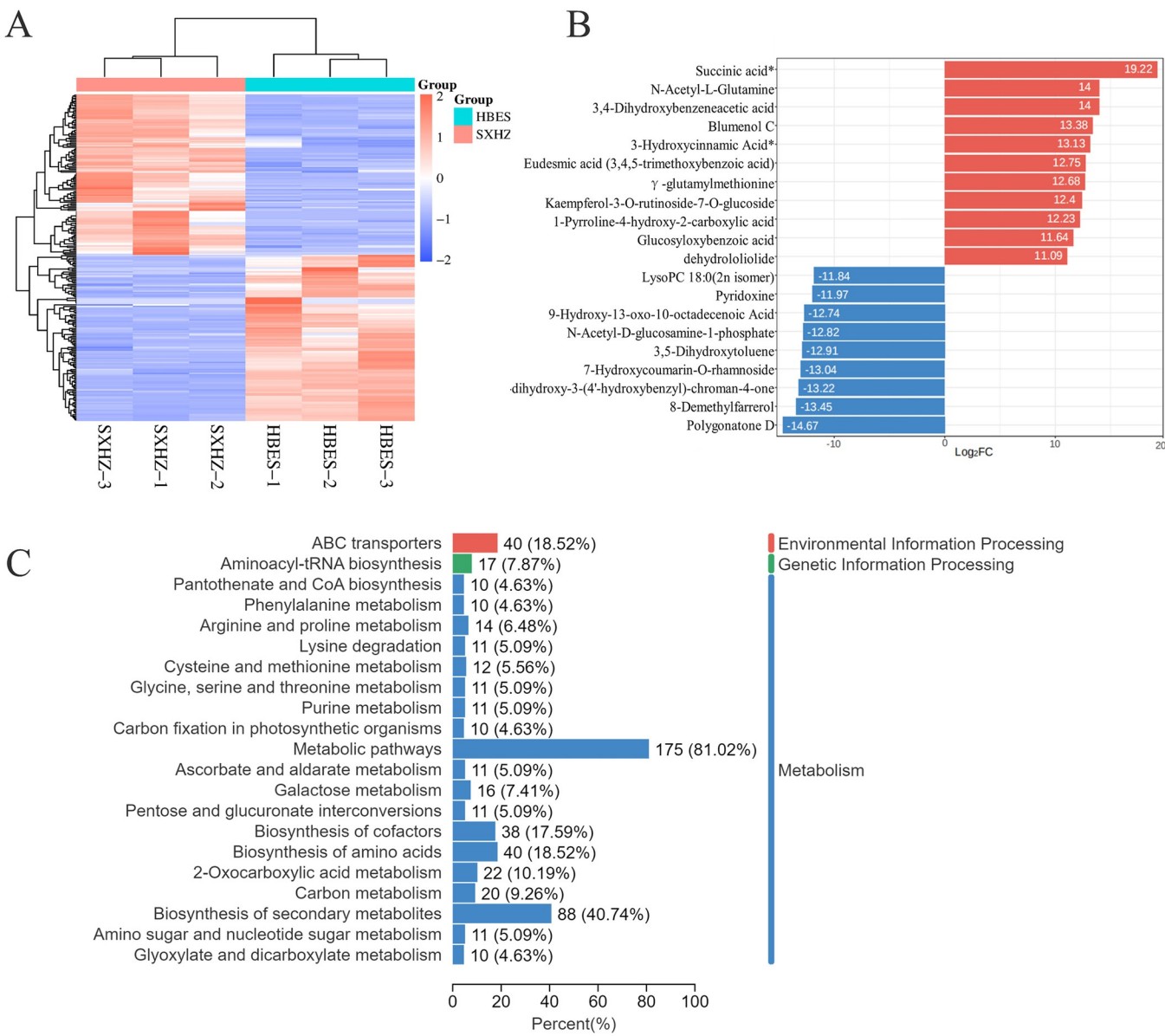

**Fig 2. Analysis of DEMs and KEGG statistics.** A: DEMs of the overall sample. B. Metabolites in the top 20 multiplicity of differences in the comparative group. C. KEGG pathway annotations of the DEMs.

In order to acquire a more comprehensive comprehension of the alterations in flavonoid metabolites in the rhizomes of two *Polygonatum* species, we conducted an investigation on the DEMs that are involved in the production and metabolism of flavonoids. As a result, we identified a total of seven DEMs. However, only two of these DEMs exhibited an increase in activity in three metabolic pathways. Apigenin-7-O-neohesperidoside (Rhoifolin) was the only DEMs found in the Flavone and flavonoid biosynthesis pathway (ko00944). Phloretin was present in both the Flavonoid biosynthesis pathways (ko00941) and secondary metabolite biosynthesis pathways (ko01110). However, Disporopsin, Apigenin-7-O-(6"-p-Coumaryl) glucoside, Chrysoeriol-7-O-rutinoside-5-O-glucoside, 8-Demethylfarrerol, and Kaempferol-3-O-rutinoside-7-O-glucoside were not found in these specific metabolic pathways.

## Sequencing and functional gene annotation

Prior to completing the bioinformatics analysis, the raw data underwent screening to confirm its high quality. By employing fastp for first read quality control, we successfully removed low-quality data, resulting in a total of 92,858 unigenes. The cumulative count of constructed nucleotide bases was 113,895,648, with a N50 value of 1,767, a N90 value of 548, and an average length of 1,227.

By conducting the sequencing of *Polygonatum* RNA, we have successfully found a total of 92,858 unigenes. The transcriptome sequencing data of the two *Polygonatum* rhizomes were annotated using seven databases. The Nr database annotated 54,381 unigenes, the KEGG database annotated 40,481 unigenes, the KOG database annotated 32,775 unigenes, and the SwissProt database annotated 38,662 unigenes. In addition, the GO database provided annotations for 46,537 unigenes, the TrEMBL database provided annotations for 54,255 unigenes, and the Pfam database provided annotations for 35,490 unigenes (Fig 3A). The KOG annotations classified a total of 32,775 genes into 25 distinct functional categories. The category with the highest number of genes was "general function prediction" (R), which had 8,030 genes (24.50%). The category with the second highest number of genes was "protein post-transporter modification, turnover, chaperone proteins" (O), which had 3,663 genes (11.18%). The category with the lowest number of genes was "cellular motility" (N), with only 12 genes (0.03%) (Fig 3B). Within the category of metabolic pathways in KEGG annotations, the annotation "Metabolic pathways" had the highest prevalence, with 2,529 genes (30.99%) (Fig 3C). Furthermore, a total of 46,537 genes were categorized into three distinct functional groups according to the Gene Ontology (GO) classification system. The category with the highest number of subcategories in Biological Processes (BP) was "cellular process", but in Cellular Components (CC), the category with the highest number of subcategories was "cellular anatomical entity". The category with the highest representation in the Molecular Function (MF) category was "Binding" (Fig 3D).

## Analysis of DEGs between groups

The analysis of differential expression identified a total of 21,553 DEGs. Comparing to the control group HBES, a total of 10418 DEGs were found to be up-regulated, while 11135 DEGs were down-regulated. The genes were subjected to GO functional analysis, which revealed category annotations and identified significant GO functional enrichment. Out of the top 20 DEGs, 14 were found to be considerably enriched. The bioprocess category showed the highest level of enrichment, as shown in Fig 5A. The Flavonoid Biosynthesis Pathway (ko00941) had 86 background genes, 30 of which were DEGs, or 34.88%, according to KEGG pathway analysis. Of the 33 background genes in the isoflavonoid biosynthesis pathway (ko00943), 14 were DEGs, or 42.42%. Of the 444 background genes in the phenylpropanoid biosynthesis pathway (ko00940), 150 were DEGs, or 33.78%. Furthermore, 3 of the 15 background genes in the flavonoid and flavonol biosynthesis pathway (ko00944) were DEGs, making up 20.00% of the total (Fig 5D).

## Gene co-expression and network analysis

Weighted Gene Co-Expression Network Analysis (WGCNA) generates a dendrogram by analyzing the relationships between gene expressions, which allows for the identification of separate modules. The merging of modules occurs when the value reaches or exceeds 0.25, and each module must contain at least 50 genes to be considered. As shown in Fig 4A, each color represents a module in the clustering tree, and each gene is allocated to a certain module. Genes that exhibit comparable changes in expression during physiological processes or across

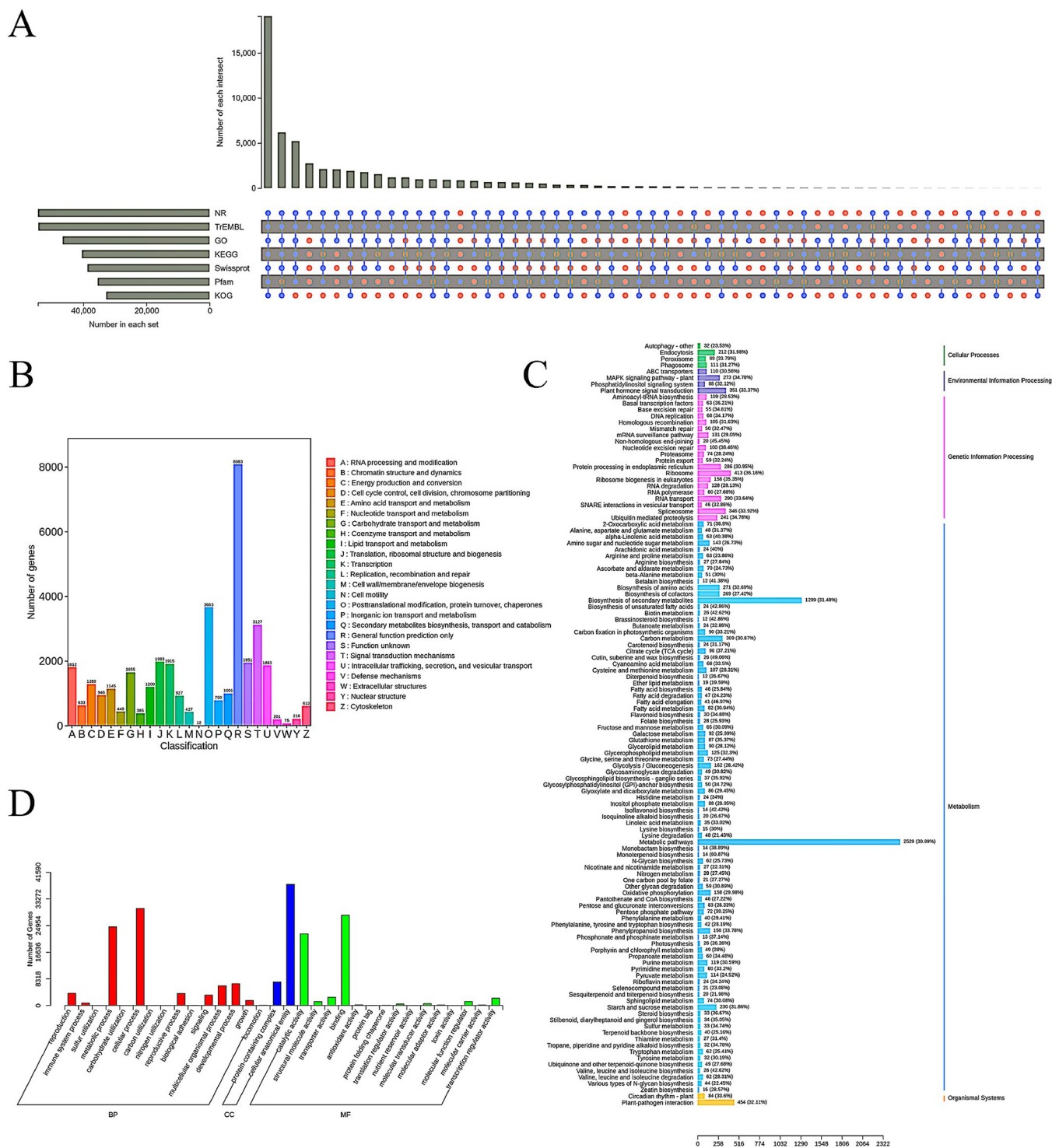

**Fig 3. Map of annotation results of two *Polygonatum* gene databases.** A: Venn diagram of 7 major database annotations. B: KOG database annotations. C: KEGG database annotations. D: GO database annotations.

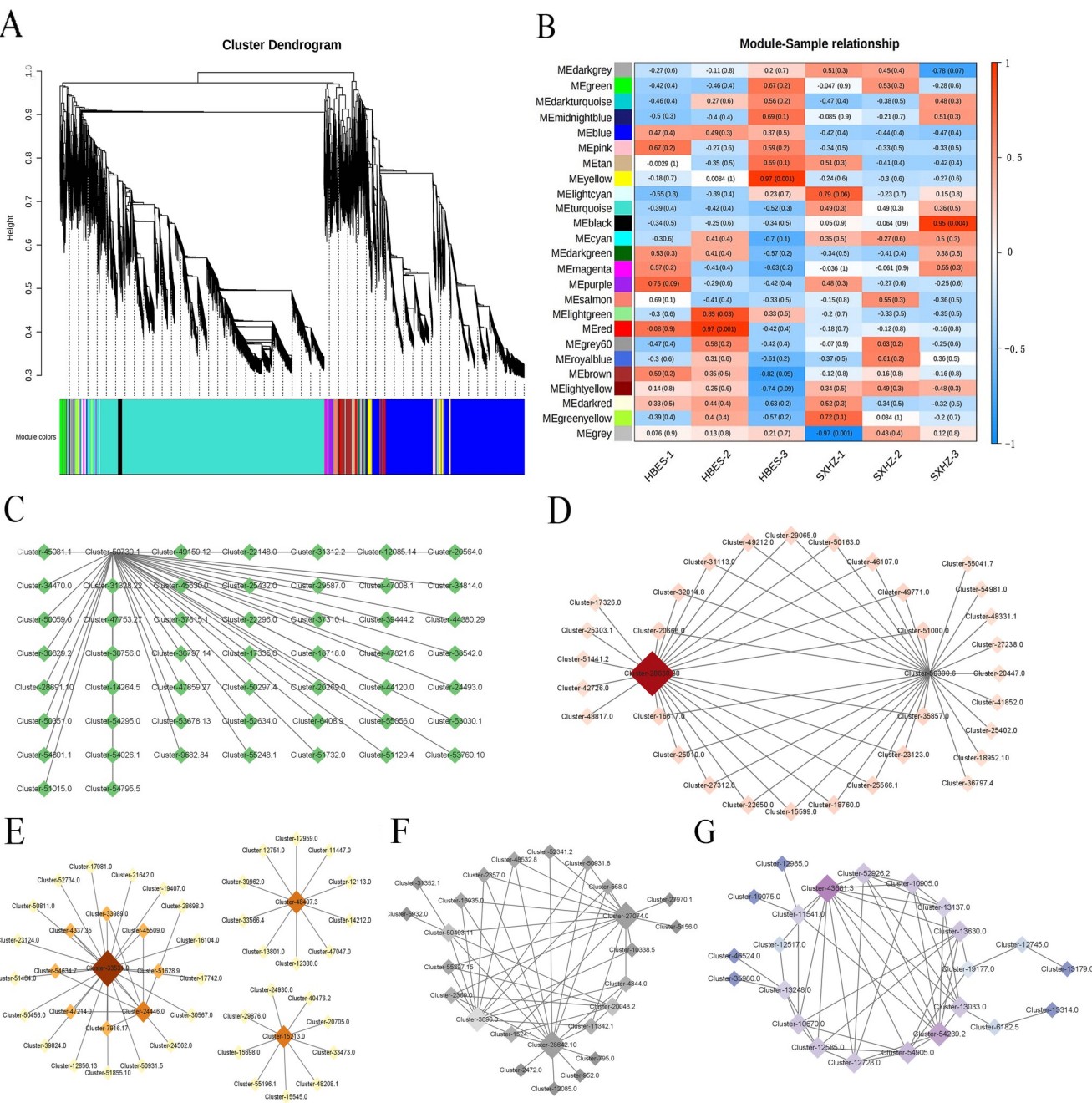

**Fig 4. Co-expression profiles of all transcripts and active components in two *Polygonatum* rhizomes.** A: Hierarchical clustering tree of the co-expressed modules of two *Polygonatum* species. B: Heat map of the correlation of the two *Polygonatum* samples and modules. C: Network diagram depicting the top 50 ranked linkage strengths of genes in the MElightgreen module. D: Network diagram illustrating the top 50 ranked linkage strengths of genes in the MEred module. E: Network diagram displaying the top 50 ranked linkage strengths of genes in the MEyellow module. F: Network diagram showing the top 50 ranked linkage strengths of genes in the MEblack module. G: Network diagram presenting the top 50 ranked linkage strengths of genes in the MEgrey module.

various tissues may be functionally linked, therefore leading to their classification within a module. In the upper portion of the tree diagram, the vertical axis reflects the genetic distance between two nodes (genes), while the horizontal axis does not have any particular significance. We divided all transcripts into 25 distinct modules. The expression levels of two *Polygonatum*

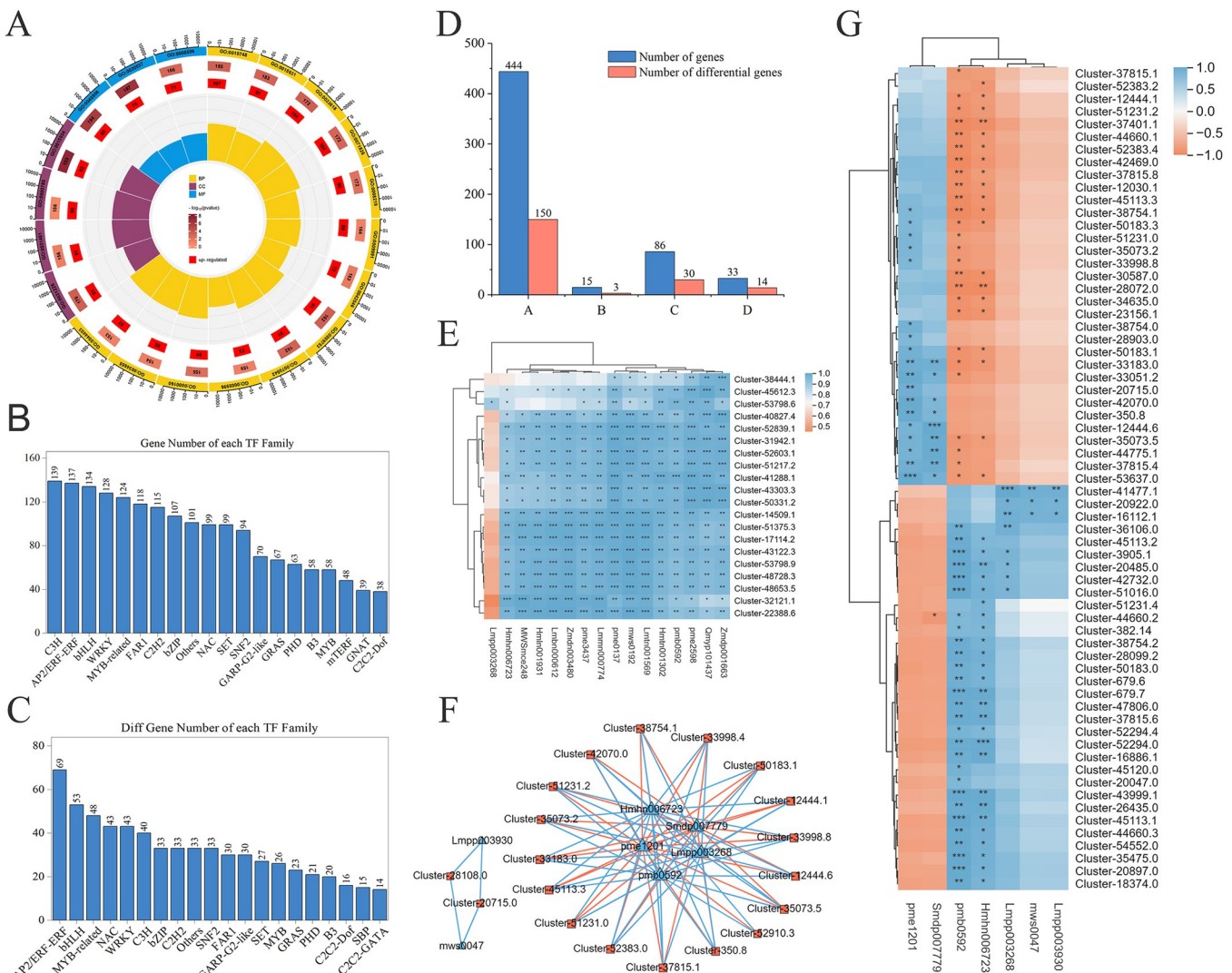

**Fig 5. Differential gene and differential metabolite analysis.** A: Enrichment circle plot of differentially expressed genes in the GO database for *Polygonatum*, the bar graph shows the proportion of up-regulated genes, and the red color represents the number of up-regulated genes. B: Number of genes in the top 20 enriched TFs family genes. C: Number of genes in the differentially expressed TFs family genes. D: Number of total genes in the KEGG pathway and the number of DEGs (the abscissa A in the middle of the pathway represents Phenylpropanoid biosynthesis; B represents Flavone and flavonol biosynthesis; C represents Flavonoid biosynthesis; D represents Isoflavonoid biosynthesis). E: FC between transcriptome and metabolome correlation heatmap of the top 20 enriched DEGs and DEGs. F: Correlation network of the top 20 DEGs and all DEMs associated with the flavonoid biosynthesis pathway. G: DEGs and DEMs involved in the flavonoid biosynthesis pathway.

genes in the MEyellow, MElightgreen, MEred, and MEblack modules showed a positive correlation, while the expression levels of genes in the MEgrey module displayed a negative correlation (P < 0.05, Fig 4B).

We investigated gene co-expression networks in five modules with positively and negatively correlated gene expression levels and analyzed the top 50 genes in each module in terms of gene linkage strength. We next examined the top 50 genes within each module based on their gene linkage strength. Within the co-expression network of the MEyellow module, Cluster-33531.0 had a high association with 25 genes. This was followed by Cluster-24446.0, Cluster-48497.3, and Cluster-15213.0 (Fig 4E). Cluster-28642.10 and Cluster-27074.0 in the MEblack module's co-expression network had strong associations with several genes (Fig 4F). In the co-

expression networks of the MEred (Fig 4D), MElightgreen (Fig 4C), and MEgrey (Fig 4G) modules, Cluster-28630.38, Cluster-50730.1, and Cluster-43681.3 were tightly related to a plurality of genes in each module, respectively.

## TFs analysis

A total of 2880 transcription factors (TFs) were annotated in the two *Polygonatum* rhizomes, which belonged to 90 TFs families (S2 Table). The C3H TFs family has the highest number of members, with 139 unigenes, followed by the AP2/ERF-ERF TFs family with 137 unigenes, and the bHLH TFs family with 134 unigenes (Fig 5B). In addition, our analysis discovered that the 978 TFs that were expressed differently could be classified into 81 families of TFs. The top 20 differentially expressed TFs families among these families were AP2/ERF-ERF, bHLH, MYB-related, NAC, WRKY, C3H, and bZIP-related TFs families (Fig 5C and S3 Table).

## Joint analysis of the transcriptome and metabolome

**Fig 5E** illustrates the heatmap of correlation enrichment for the top 20 DEMs and DEGs, ranked by FC between the transcriptome and metabolome. The color variances represent distinct correlation coefficients. The correlation network of the top 20 DEGs and all DEMs linked to the flavonoid biosynthesis pathway was also mapped (Fig 5F). A strong link was found among Lmpp003268, pmb0592, Hmhn006723, pme1201, Smdp007779, and 17 additional DEGs in this network. In addition, mws0047 and Lmpp003930 showed a stronger connection with Cluster-28108.0 and Cluster-20715.0, respectively. The enrichment heatmap displays the DEGs and DEMs that play a role in the production pathway of flavonoids (Fig 5G).

Four metabolites were discovered in the investigation of flavonoid production routes: phloretin, caffeoyl quinic acid, neohesperidin, and (-)-Epiafzelechin (Fig 6). However, only Phloretin showed substantial variations that were distinct from the others. The pathway analysis identified the participation of 32 distinct enzymes that have a significant impact on 30 genes. After de-emphasis, the enzymes that are included are: anthocyanidin reductase (ANR), dihydroflavonol 4-reductase (DFR), flavanone 4-reductase (FNR), 5-O-(4-coumaroyl)-D-quinate 3'- monooxygenase (C3'H), caffeoyl-CoA O-methyltransferase (CCoAOMT), chalcone isomerase (CHI), flavonoid 3'-monooxygenase (F3'H), flavonol synthase (FLS), shikimate O-hydroxycinnamoyltransferase (HCT), and phlorizin synthase (PHZS) as shown in S4 Table. In addition, out of the 30 genes, five were strongly linked to PHZS and all of them belonged to the MYB-related TFs family.

## Validation of DEGs by qRT–PCR

In order to validate the precision of the transcriptome data, we conducted RNA-Seq analysis and reverse transcription PCR (qRT-PCR) on a randomly chosen set of differentially expressed genes (DEGs) to ascertain the genuineness and dependability of the transcriptome data. We chose 20 genes that showed differential expression for validation using qRT-PCR. While there were minor discrepancies between the results of RNA-Seq and qRT-PCR, the overall trend remained consistent for around 80% of the genes (Fig 7). Thus, transcriptome sequencing is deemed reliable.

## Discussion

Flavonoids are present in numerous plants and possess a wide range of characteristics, such as antioxidant, anti-aging, antiviral, and anticancer capabilities [25]. Prior research has established that *Polygonatum* possesses a significant abundance of flavonoids, which are widely

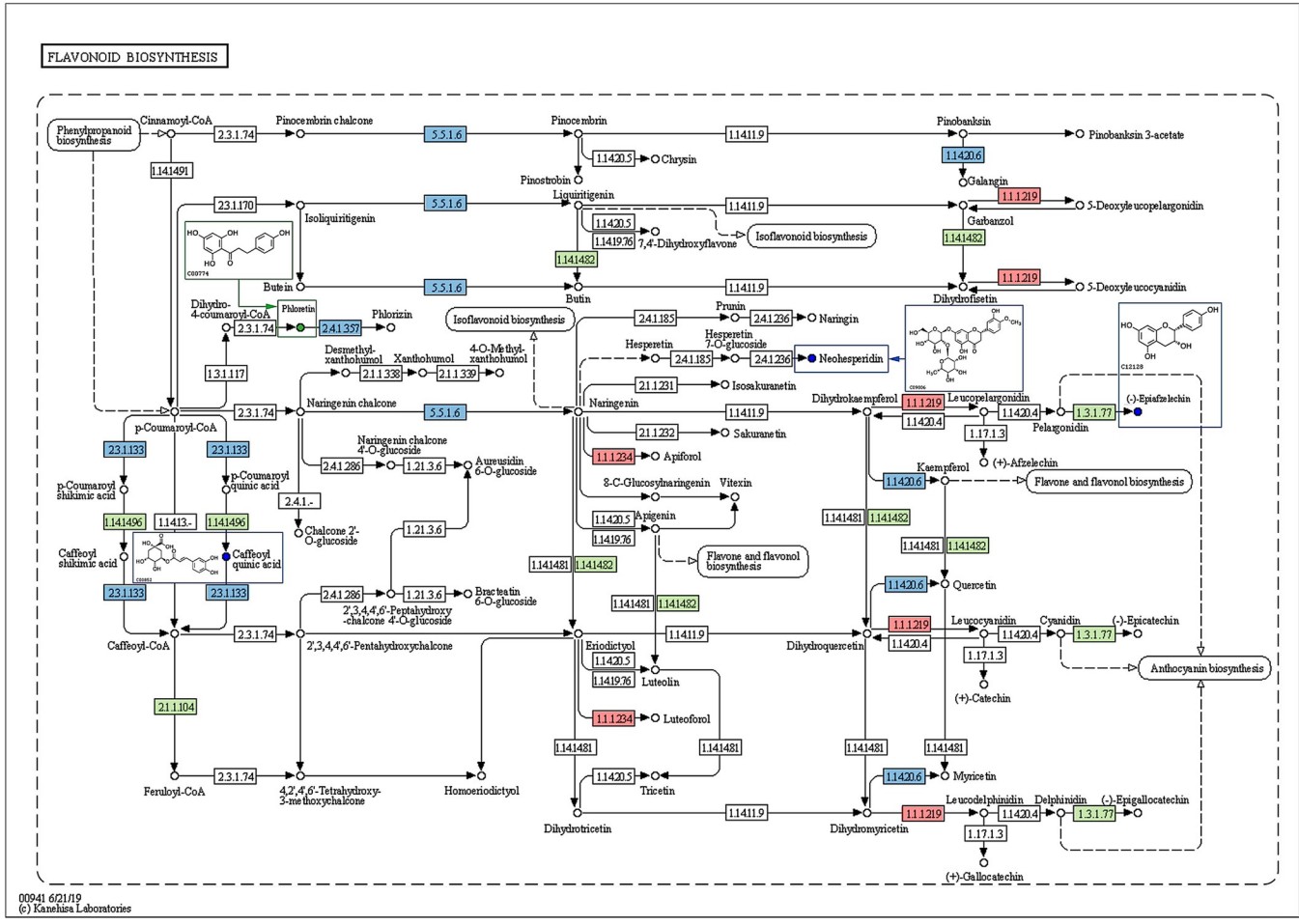

**Fig 6. Study of flavonoid biosynthesis and metabolic pathways in *Polygonatum* rhizomes.** Blue dots indicate that the metabolite was detected but not significantly changed, and green dots indicate that the metabolite content was significantly down-regulated in the experimental group. Red rectangles indicate that the enzyme is associated with up-regulated genes, green rectangles indicate that the enzyme is associated with down-regulated genes, blue rectangles indicate that the enzyme is associated with both up- and down-regulated genes, and the numbers in the boxes represent the enzyme numbers (EC numbers).

recognized for their substantial medicinal benefits [26]. Hence, the examination of the mechanisms governing flavonoid synthesis and strategies to enhance their abundance has emerged as a critical area of research. The objective of this study was to investigate the process of flavonoid production in HBES and SXHZ by analyzing their metabolomic and transcriptomic data. This technique enhanced our comprehension of the regulatory network, the accumulation of flavonoids, and the molecular mechanisms that propel the development of *Polygonatum* rhizomes. These findings offer valuable insights for future research in this field.

All transcripts in this study can be divided into 25 distinct modules. The expression levels of two *Polygonatum* genes in the MEyellow, MElightgreen, MEred, and MEblack modules were positively correlated, while the gene expression levels in the MEgrey module were negatively correlated. A co-expression analysis was conducted on the top 50 gene pairings that had the strongest association within these five modules. Genes such as Cluster-33531.0, Cluster-15213.0, Cluster-28642.10, Cluster-27074.0, Cluster-28630.38, Cluster-50730.1, and Cluster-43681.3 had robust associations with several genes inside their respective modules. The construction of co-expression networks unveiled a multitude of pivotal genes that perform vital roles in *Polygonatum*. However, the genes depicted in the figure did not include any

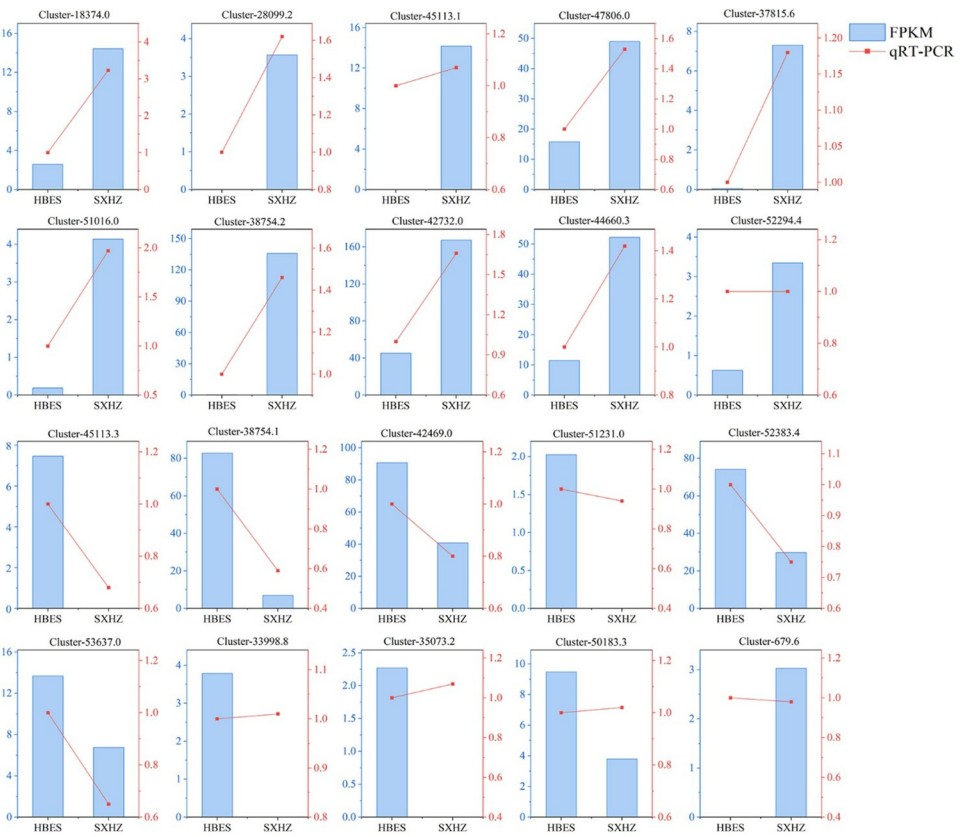

**Fig 7. Results of qRT-PCR assay.** The bar graph represents the FPKM results of transcriptome data; the line graph represents the qRT-PCR results.

differentially expressed genes linked to flavonoid production, indicating that further analysis of these genes is necessary.

The bHLH, WRKY, AP2, MBW-related, C3H, and other TF families showed significant enrichment and differential expression in this study. Additionally, 30 genes related to the flavonoid biosynthesis pathway exhibited significant differential expression, with five closely related to PHZS. PHZS is a member of the MYB-related transcription factor family. The above results suggest that the MYB-related family may be mainly responsible for controlling genes in *Polygonatum*'s flavonoid biosynthesis pathway. The TF family is a transcription factor group with similar structures and functions. They control the activity of target genes by binding to specific DNA sequences and are very important in controlling gene expression [27]. So far, regulatory genes that directly impact the flavonoid biosynthesis pathway have been discovered in several plant species. These families have attracted considerable attention because they can function autonomously or form MBW complexes to control the structural genes involved in the flavonoid biosynthesis pathway [28, 29]. The PbUFGT1 promoter, derived from the MYB TFs family, can stimulate the synthesis of anthocyanins and flavonols in pear fruits [30]. Meanwhile, the upregulation of the CcMYB12 gene from the MYB TTF family induces a rise in flavonol levels in Arabidopsis leaves [31]. Additionally, the WRKY and AP2 families of TFs have essential functions in producing flavonoid molecules [32]. In summary, the differences in transcription factors are one of the crucial reasons affecting the biosynthesis of the two flavonoid compounds in *Polygonatum* rhizome.

Metabolomics analysis showed that HBES and SXHZ contained a total of 71 flavonoids. Among them, seven were identified as DEMs. When comparing the SXHZ to the control (HBES), it was found that the SXHZ had a significantly higher amount of the kaempferol derivative kaempferol-3-O-rutinoside-7-O-glucoside. The difference in content between the two was as high as 5390.82-fold. There were significant increases observed in other flavonoids, such as chrysoeriol-7-O-rutinoside-5-O-glucoside, which showed a fold rise of 1147.05. Disporopsin had a fold increase of 920.64, Apigenin-7-O-neohesperidoside had a fold increase of 62.61, and apigenin-7-O-(6"-p-Coumaryl) glucoside had a fold increase of 52.35. Previous studies have demonstrated that kaempferol and its derivatives exhibit neuroprotective effects and are highly active flavonoids [33]. Chrysoeriol has been discovered to hinder the accumulation of fat in adipocytes [34]. Apigenin and its derivatives have demonstrated a variety of pharmacological actions, such as anti-inflammatory, anticancer, antioxidant, antiradiation, antidepressant, cardiovascular protection, modulation of glycolipid metabolism, nephroprotection, neuroprotection, and antibacterial characteristics [35]. In the present study, we found significant differences in the flavonoid composition of HBES and SXHZ. This implies that by the cultivation of several types of *Polygonatum*, it could be feasible to get distinct flavonoids. This finding offers the possibility of choosing *Polygonatum* herbs that contain certain flavonoids for the treatment of different ailments.

Within the flavonoid biosynthesis route, only the phloretin content exhibited a notable reduction, with SXHZ being 1137.28 times greater than HBES. The production of phloretin in this pathway is primarily regulated by enzymes such as chalcone synthase (CHS) and hydroxycinnamoyl-CoA reductase (HCR). Nevertheless, CHS and HCR were not discovered. Phenylalanine ammonia-lyase (PAL), an enzyme in the phenylpropanoid biosynthesis pathway, was notably increased, which could potentially change the levels of p-coumaroyl-CoA and dihydro-4-coumaroyl-CoA. This change indirectly impacts the amount of phloretin present. Therefore, PAL could potentially have a significant role in the variation of phloretin levels seen between SXHZ and HBES. In addition, ANR, CCoAOMT, F3'H, and C3'H enzymes were found to be connected with genes that were downregulated, whereas FNR and DFR enzymes were linked to genes that were upregulated. CHI, FLS, PHZS, and HCT were linked to genes that were both upregulated and downregulated. Prior research has demonstrated that ANR plays a crucial role as an enzyme in the process of proanthocyanidin production [36]. D-caffeoylquinate undergoes conversion by C3'H enzymes to become 5-O-(4-coumaroyl)-D-quinate, which is a precursor of flavonoids. The intermediate compound undergoes additional reactions facilitated by enzymes, resulting in the production of other types of flavonoids, including quercetin and quercetin-3-O-glucoside. Therefore, C3'H plays a crucial function in the production route of flavonoids and enhances the biosynthesis of these molecules [37]. F3'H enzyme facilitates the process of hydroxylating dihydroflavonols and flavonols, resulting in the breakdown of kaempferol. On the other hand, FLS enzyme facilitates the conversion of dihydroflavonols into flavonols, thereby enhancing the synthesis of kaempferol [38]. Multiple studies have emphasized the importance of CCoAOMT, CHI, FNR, and DFR as pivotal enzymes in the process of flavonoid production [39–42]. Thus, we propose that notable alterations in the expression of genes such as ANR, CCoAOMT, F3'H, C3'H, CHI, FLS, PHZS, and HCT may also play a role in the variations seen in the flavonoid production pathways between the two *Polygonatum* rhizomes.

## Conclusion

A total of 71 flavonoids were detected in the metabolomic data analysis, with noticeable variations found in the content of seven. Apigenin exhibited considerable differential expression in

the flavonoid biosynthesis pathway within the DEMs. Furthermore, the transcription factor families bHLH, WRKY, AP2, MBW-related, and C3H exhibited a high level of enrichment in the metabolomic data, and these changes were statistically significant. The DEGs in the flavonoid biosynthesis pathway primarily belonged to the family of TFs associated to the MBW complex. This suggests that the family of TFs connected to MYB might have a crucial function in regulating the flavonoid biosynthesis pathway in *Polygonatum* rhizomes. Meanwhile, alterations in DEGs such as PAL, ANR, CCoAOMT, F3'H, C3'H, CHI, FLS, PHZS, and HCT may have a role in the variations seen in the flavonoid biosynthesis pathway between the two *Polygonatum* rhizomes. Taken together, these findings provide new insights into understanding the biosynthesis and accumulation of flavonoid compounds in *Polygonatum* rhizomes. The differential accumulation of different compounds may affect the medicinal value of different *Polygonatum* species. Therefore, we suggest further in-depth exploration of the metabolism and regulatory mechanisms of flavonoids in *Polygonatum*, aiming to guide the cultivation of new high-quality *Polygonatum* varieties with high flavonoid content.

## Supporting information

**S1 Table. qRT-PCR primer list.**
(XLSX)

**S2 Table. Summary table of transcription factor families.**
(XLSX)

**S3 Table. Top 20 most significantly differentially expressed TFs families.**
(XLSX)

**S4 Table. Differential genes in the flavonoid biosynthesis pathway.**
(XLSX)

## Author Contributions

**Conceptualization:** Xiaolin Wan.

**Data curation:** Xiaolin Wan, Qiang Xiao.

**Investigation:** Xiaolin Wan.

**Methodology:** Xiaolin Wan.

**Project administration:** Qiang Xiao.

**Validation:** Xiaolin Wan.

**Visualization:** Xiaolin Wan.

**Writing – original draft:** Xiaolin Wan.

**Writing – review & editing:** Qiang Xiao.

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
