## [Decision Letter · Decision Letter 0]

23 Oct 2024

PONE-D-24-28928Integrative analysis of the metabolome and transcriptome provides insights into the mechanisms of flavonoid biosynthesis in PolygonatumPLOS ONE

Dear Dr. Xiao,

Thank you for submitting your manuscript to PLOS ONE. After careful consideration, we feel that it has merit but does not fully meet PLOS ONE’s publication criteria as it currently stands. Therefore, we invite you to submit a revised version of the manuscript that addresses the points raised during the review process.

We look forward to receiving your revised manuscript.

Kind regards,

Vikas Sharma, Ph.D

Academic Editor

PLOS ONE

Journal Requirements:

2. Thank you for submitting the above manuscript to PLOS ONE. During our internal evaluation of the manuscript, we found significant text overlap between your submission and previous work in the [introduction, conclusion, etc.].

Please revise the manuscript to rephrase the duplicated text, cite your sources, and provide details as to how the current manuscript advances on previous work. Please note that further consideration is dependent on the submission of a manuscript that addresses these concerns about the overlap in text with published work.

[If the overlap is with the authors’ own works: Moreover, upon submission, authors must confirm that the manuscript, or any related manuscript, is not currently under consideration or accepted elsewhere. If related work has been submitted to PLOS ONE or elsewhere, authors must include a copy with the submitted article. Reviewers will be asked to comment on the overlap between related submissions (http://journals.plos.org/plosone/s/submission-guidelines#loc-related-manuscripts).]

We will carefully review your manuscript upon resubmission and further consideration of the manuscript is dependent on the text overlap being addressed in full. Please ensure that your revision is thorough as failure to address the concerns to our satisfaction may result in your submission not being considered further.

Additional Editor Comments:

Author's are required to address the issues raised by reviewers carefully otherwise manuscript will not be considered further.

Comments from the Journal Office:

One or more of the reviewers has recommended that you cite specific previously published works. Members of the editorial team have determined that the works referenced are not directly related to the submitted manuscript. As such, please note that it is not necessary or expected to cite the works requested by the reviewer.

Reviewers' comments:

Reviewer's Responses to Questions

Comments to the Author

1. Is the manuscript technically sound, and do the data support the conclusions?

Reviewer #1: Yes

Reviewer #2: Yes

2. Has the statistical analysis been performed appropriately and rigorously? 

Reviewer #1: Yes

Reviewer #2: Yes

3. Have the authors made all data underlying the findings in their manuscript fully available?

Reviewer #1: Yes

Reviewer #2: Yes

4. Is the manuscript presented in an intelligible fashion and written in standard English?

Reviewer #1: Yes

Reviewer #2: Yes

5. Review Comments to the Author

Reviewer #1: Introduction

The description of the benefits and uses of Polygonatum plants has some repetitive information, especially in the sections discussing their bioactive compounds and flavonoid-related properties. Consider consolidating these sections to avoid redundancy.

- The introduction seems disjointed, jumping between topics such as medicinal uses, bioactive compounds, and transcription factors without clear transitions. It would be helpful to organize the content more logically. You might start with the general importance of Polygonatum, then focus specifically on flavonoids, and finally transition into the role of transcription factors in flavonoid biosynthesis.

- Some sentences are overly long and complex, which can obscure the main point. For instance, consider breaking the sentence starting with "In previous studies, chalcone synthase is the entry-step enzyme..." into shorter, more precise sentences.

Methodology

While the methodology provides sufficient detail for someone experienced in the field, certain sections could benefit from more explicit instructions to ensure reproducibility. For example, when discussing RNA sequencing, it would be helpful to specify how contaminants were removed and how the quality of the RNA was assessed before sequencing.

The description of statistical analysis is brief. For instance, in the qRT-PCR section, the use of the 2-ΔΔCt method is mentioned. Still, there's no discussion of how data were normalized (e.g., reference genes used) or whether statistical tests were applied to determine significance."

Results

Despite efforts to validate the multivariate models, the risk of overfitting remains a concern, especially when dealing with high-dimensional data such as metabolomics. The high R2Y and Q2 values (0.998 and 0.935, respectively) indicate that the model fits the training data exceptionally well, but this does not necessarily guarantee good predictive performance on new data.

The study reports a large amount of transcriptomic data, with over 92,000 unigenes identified. However, the integration of these data with the metabolomic findings is not as strong as it could be. More effort could be made to link specific gene expression patterns with the metabolite profiles observed, especially in the context of the metabolic pathways identified. This could help in understanding the regulatory networks governing metabolite production in Polygonatum species.

The analysis of co-expression profiles and transcription factors is comprehensive, but there is a lack of discussion regarding the biological significance of these findings. For example, while the C3H, AP2/ERF-ERF, and bHLH families are identified as the most prevalent TFs, the study does not elaborate on their specific roles in Polygonatum rhizomes.

The study attempts to integrate transcriptome and metabolome data, but the connection between the two datasets could be more robust. The heatmap in Fig. 5E and the correlation network in Fig. 5F are visually informative, but they lack sufficient interpretation. For example, the identification of four metabolites (Phloretin, Caffeoyl quinic acid, Neohesperidin, and (-)-Epiafzelechin) is noteworthy, but the study should delve deeper into how these metabolites and their associated DEGs contribute to the overall metabolic pathways and biological functions of the plant.

Reviewer #2: Author are suggested to improve MS.

1.Author must include replication details in transcriptome study in material method

2.Transcriptome result must be fortified with biology. Only writing bioinformatics and few numbers are not good.Pleas refer following manuscript to understand.

https://www.nature.com/articles/srep30412

3.material method are repeated in results. Author are suggested to write about only results in results section instead of name of many tools and software

4.Out of these15333 modules, MEyellow, MElightgreen, MEred, and MEblack displayed a positive link with the active

334 components, whereas the MEgrey module demonstrated a negative association. " This sentence missing biology

5.Discussion is very poor.

Author are strongly suggested to discuss result in discussion instead of any theory. Please avoid to discuss software and techniqyes in result and discussion.You are strongly suggested to refer

https://www.nature.com/articles/srep30412

6. PLOS authors have the option to publish the peer review history of their article (what does this mean?). If published, this will include your full peer review and any attached files.

Do you want your identity to be public for this peer review? For information about this choice, including consent withdrawal, please see our Privacy Policy.

Reviewer #1: No

Reviewer #2: Yes: Kuldip Jayaswall

---

## [Author Response · Author response to Decision Letter 0]

25 Oct 2024

Dear Editor,

Thank you very much for kindly providing us with the opportunity to revise our manuscript entitled " Integrative analysis of the metabolome and transcriptome provides insights into the mechanisms of flavonoid biosynthesis in Polygonatum ". The reviewers’ comments are all valuable and very helpful for revising and improving our paper. The revised sections are highlighted in the revision. The main corrections in the paper and the response to the reviewer’s comments are as follows:

Response to Reviewers:

Reviewer #1:

Introduction

1. The description of the benefits and uses of Polygonatum plants has some repetitive information, especially in the sections discussing their bioactive compounds and flavonoid-related properties. Consider consolidating these sections to avoid redundancy. 

Response: Thank you very much for reviewing our research and providing valuable feedback. We have integrated the redundant parts describing the effects and uses of the Polygonatum plant in the manuscript. Please refer to lines 46-60 of the revised draft for details.

2. The introduction seems disjointed, jumping between topics such as medicinal uses, bioactive compounds, and transcription factors without clear transitions. It would be helpful to organize the content more logically. You might start with the general importance of Polygonatum, then focus specifically on flavonoids, and finally transition into the role of transcription factors in flavonoid biosynthesis. 

Response: Thank you very much for reviewing our research and providing valuable feedback. We have made every effort to revise the introduction to make it more logical. However, our revisions may still have shortcomings. If you have any further questions, please feel free to contact us, and we will respond as soon as possible.

3. Some sentences are overly long and complex, which can obscure the main point. For instance, consider breaking the sentence starting with "In previous studies, chalcone synthase is the entry-step enzyme..." into shorter, more precise sentences. 

Response: Thank you very much for reviewing our research and providing valuable feedback. We have revised the overly long and complex sentences, as can be seen in the revised version on lines 91-93. In addition, we have also made changes to other overly long and complex sentences in the introduction section of the manuscript, which can be found in the revised version on lines 46-123.

Methodology

1. While the methodology provides sufficient detail for someone experienced in the field, certain sections could benefit from more explicit instructions to ensure reproducibility. For example, when discussing RNA sequencing, it would be helpful to specify how contaminants were removed and how the quality of the RNA was assessed before sequencing. 

Response: Thank you very much for reviewing our research and providing valuable feedback. We have specifically explained how to remove contaminants and how to assess RNA quality before sequencing. Please refer to lines 155-162 of the revised manuscript for details.

2. The description of statistical analysis is brief. For instance, in the qRT-PCR section, the use of the 2-ΔΔCt method is mentioned. Still, there's no discussion of how data were normalized (e.g., reference genes used) or whether statistical tests were applied to determine significance." 

Response: Thank you very much for reviewing our research and providing valuable feedback. We have included the reference gene sequences used and conducted relative quantification strictly following the 2-ΔΔCt method, with the normalization also based on the 2-ΔΔCt method. Please see the revised version on lines 199-202.

Results

1. Despite efforts to validate the multivariate models, the risk of overfitting remains a concern, especially when dealing with high-dimensional data such as metabolomics. The high R2Y and Q2 values (0.998 and 0.935, respectively) indicate that the model fits the training data exceptionally well, but this does not necessarily guarantee good predictive performance on new data. 

Response: Thank you very much for reviewing our research and providing valuable feedback. We have reviewed numerous studies and found that many have used the OPLS-DA model (R2X, R2Y, and Q2) for data prediction, with results showing that the model performs satisfactorily on new data. Therefore, we believe that the current model evaluation methods and results sufficiently support the conclusions of this paper. If you have any further suggestions, please feel free to contact us. We will respond to you as soon as possible.

2. The study reports a large amount of transcriptomic data, with over 92,000 unigenes identified. However, the integration of these data with the metabolomic findings is not as strong as it could be. More effort could be made to link specific gene expression patterns with the metabolite profiles observed, especially in the context of the metabolic pathways identified. This could help in understanding the regulatory networks governing metabolite production in Polygonatum species. 

Response: Thank you very much for reviewing our research and providing valuable feedback. In the manuscript, we conducted a joint analysis of the metabolome and transcriptome, and illustrated the flavonoid biosynthesis pathway in Figure 6. Through Figure 6, we can clearly observe the upregulation and downregulation of metabolites and genes in the flavonoid biosynthesis pathway in the two Polygonatum rhizomes. Please refer to lines 325-343 of the manuscript for more details.

3. The analysis of co-expression profiles and transcription factors is comprehensive, but there is a lack of discussion regarding the biological significance of these findings. For example, while the C3H, AP2/ERF-ERF, and bHLH families are identified as the most prevalent TFs, the study does not elaborate on their specific roles in Polygonatum rhizomes. 

Response: Thank you very much for reviewing our research and providing valuable feedback. In the manuscript, we only analyzed the differences in genes and metabolites between the two Polygonatum species, aiming to better distinguish between them and provide a theoretical basis for the future development of the Polygonatum industry. The gene families we identified in Polygonatum require further functional validation in future studies, which is why we did not elaborate on their roles in detail. In the future, we will conduct more in-depth research focusing on these gene families.

4. The study attempts to integrate transcriptome and metabolome data, but the connection between the two datasets could be more robust. The heatmap in Fig. 5E and the correlation network in Fig. 5F are visually informative, but they lack sufficient interpretation. For example, the identification of four metabolites (Phloretin, Caffeoyl quinic acid, Neohesperidin, and (-)-Epiafzelechin) is noteworthy, but the study should delve deeper into how these metabolites and their associated DEGs contribute to the overall metabolic pathways and biological functions of the plant. 

Response: Thank you very much for reviewing our research and providing valuable feedback. We detected four metabolites: phenol, chlorogenic acid, neohesperidin, and (-)-epiafzelechin, but only phenol exhibited significant differential changes. Additionally, we have discussed this compound in detail in the discussion section.

Reviewer #2: 

Author are suggested to improve MS.

1.Author must include replication details in transcriptome study in material method在

Response: Thank you very much for reviewing our research and providing valuable feedback. We have added detailed information about the transcriptome sequencing, which can be found in the revised version on lines 157-162.

2.Transcriptome result must be fortified with biology. Only writing bioinformatics and few numbers are not good.Pleas refer following manuscript to understand.

https://www.nature.com/articles/srep30412

Response: Thank you very much for reviewing our research and providing valuable feedback. We carefully read the article you recommended and found that it explores candidate genes related to the defense against water bubble disease in tea plants from a transcriptomic perspective. This study closely connects bioinformatics analysis with biology. However, our paper primarily investigates the differences in flavonoid biosynthesis between the two Polygonatum species. We identified that phenol and the MYB-related transcription factor gene family play a major role in regulating the flavonoid biosynthesis pathway in Polygonatum rhizomes. We believe this finding is of significant reference value for the future breeding of Polygonatum germplasm resources with high flavonoid content and the development of the related pharmaceutical industry. In future research, we will conduct more in-depth studies on Polygonatum, including the relationship between transcriptomic results and biology.

3.material method are repeated in results. Author are suggested to write about only results in results section instead of name of many tools and software. 

Response: Thank you very much for reviewing our research and providing valuable feedback. We have checked and removed the redundant content in the Materials and Methods section and the Results section. Additionally, we have carefully revised the content of the Results section to ensure that it presents only the results.

4.Out of these15333 modules, MEyellow, MElightgreen, MEred, and MEblack displayed a positive link with the active 334 components, whereas the MEgrey module demonstrated a negative association. " This sentence missing biology

Response: Thank you very much for reviewing our research and providing valuable feedback. We have revised that sentence. Please see the revised version on lines 296-298. We may not have fully understood your question, so if you have any further inquiries, please feel free to contact us.

5.Discussion is very poor.

Author are strongly suggested to discuss result in discussion instead of any theory. Please avoid to discuss software and techniqyes in result and discussion.You are strongly suggested to refer

https://www.nature.com/articles/srep30412. 

Response: Thank you very much for reviewing our research and providing valuable feedback. We have made every effort to revise the discussion section of the article to make it more logical. Furthermore, we did not discuss software and technology in the discussion.

---

## [Decision Letter · Decision Letter 1]

8 Jan 2025

Integrative analysis of the metabolome and transcriptome provides insights into the mechanisms of flavonoid biosynthesis in Polygonatum

PONE-D-24-28928R1

Dear Dr. Xiao,

We’re pleased to inform you that your manuscript has been judged scientifically suitable for publication and will be formally accepted for publication once it meets all outstanding technical requirements.

Kind regards,

Dan Gao, Ph.D.

Academic Editor

PLOS ONE

Additional Editor Comments (optional):

Reviewers' comments:

Reviewer's Responses to Questions

**Comments to the Author**

1. If the authors have adequately addressed your comments raised in a previous round of review and you feel that this manuscript is now acceptable for publication, you may indicate that here to bypass the “Comments to the Author” section, enter your conflict of interest statement in the “Confidential to Editor” section, and submit your "Accept" recommendation.

Reviewer #1: All comments have been addressed

Reviewer #2: All comments have been addressed

2. Is the manuscript technically sound, and do the data support the conclusions?

Reviewer #1: Yes

Reviewer #2: Yes

3. Has the statistical analysis been performed appropriately and rigorously? 

Reviewer #1: Yes

Reviewer #2: Yes

4. Have the authors made all data underlying the findings in their manuscript fully available?

Reviewer #1: Yes

Reviewer #2: Yes

5. Is the manuscript presented in an intelligible fashion and written in standard English?

Reviewer #1: Yes

Reviewer #2: Yes

6. Review Comments to the Author

Reviewer #1: (No Response)

Reviewer #2: Manuscript entitled "Integrative analysis of the metabolome and transcriptome provides insights into the

mechanisms of flavonoid biosynthesis in Polygonatum" may be acceted

7. PLOS authors have the option to publish the peer review history of their article (what does this mean?). If published, this will include your full peer review and any attached files.

Reviewer #1: **Yes: **Baljinder Singh

Reviewer #2: **Yes: **Kuldip Jayaswall

---

## [Editor Report · Acceptance letter]

12 Jan 2025

PONE-D-24-28928R1 

PLOS ONE

Dear Dr. Xiao, 

I'm pleased to inform you that your manuscript has been deemed suitable for publication in PLOS ONE. Congratulations! Your manuscript is now being handed over to our production team.

Kind regards, 

on behalf of

Dr. Dan Gao 

Academic Editor

PLOS ONE